# eDNA Inactivation and Biofilm Inhibition by the PolymericBiocide Polyhexamethylene Guanidine Hydrochloride (PHMG-Cl)

**DOI:** 10.3390/ijms23020731

**Published:** 2022-01-10

**Authors:** Olena V. Moshynets, Taras P. Baranovskyi, Olga S. Iungin, Nadiia P. Kysil, Larysa O. Metelytsia, Ianina Pokholenko, Viktoria V. Potochilova, Geert Potters, Kateryna L. Rudnieva, Svitlana Y. Rymar, Ivan V. Semenyuta, Andrew J. Spiers, Oksana P. Tarasyuk, Sergiy P. Rogalsky

**Affiliations:** 1Biofilm Study Group, Department of Cell Regulatory Mechanisms, Institute of Molecular Biology and Genetics, National Academy of Sciences of Ukraine, 150 Zabolotnoho Str., 03680 Kiev, Ukraine; olgaungin@gmail.com (O.S.I.); yasnenka@gmail.com (I.P.); s.y.rymar@gmail.com (S.Y.R.); 2Department of Dermatovenerology, Allergology, Clinical and Laboratory Immunology, Shupyk National Healthcare University of Ukraine, 9 Dorohozhytska Str., 03680 Kiev, Ukraine; taras.baranovskyi@gmail.com; 3Kyiv Regional Clinical Hospital, 1 Baggovutivska Street, 04107 Kiev, Ukraine; vika.ptch@gmail.com (V.V.P.); kateryna.rudneva@gmail.com (K.L.R.); 4Department of Biotechnology, Leather and Fur, Faculty of Chemical and Biopharmaceutical Technologies, Kyiv National University of Technologies and Design, Nemyrovycha-Danchenka Street, 2, 01011 Kiev, Ukraine; 5National Children’s Specialized Hospital “Okhmatdyt”, 28/1 Chornovola Str., 01135 Kiev, Ukraine; nadejda.kisel1961@gmail.com; 6V. P. Kukhar Institute of Bioorganic Chemistry and Petrochemistry, National Academy of Science of Ukraine, 50 Kharkivske Schose, 01135 Kiev, Ukraine; LarisaMetelitsa@gmail.com (L.O.M.); ivan@bpci.kiev.ua (I.V.S.); oksanatarasyuk1201@gmail.com (O.P.T.); 7Antwerp Maritime Academy, Noordkasteel Oost 6, 2030 Antwerp, Belgium; geert.potters@hzs.be; 8Department of Bioscience Engineering, University of Antwerp, Groenenborgerlaan 171, 2020 Antwerp, Belgium; 9School of Applied Sciences, Abertay University, Bell Street, Dundee DD1 1HG, UK; a.spiers@abertay.ac.uk

**Keywords:** eDNA, antibiotic resistance, biofilms, biocides, disinfectant, alcohols, hydrogen peroxide, quaternary ammonium compounds, polyhexamethylene guanidine

## Abstract

The choice of effective biocides used for routine hospital practice should consider the role of disinfectants in the maintenance and development of local resistome and how they might affect antibiotic resistance gene transfer within the hospital microbial population. Currently, there is little understanding of how different biocides contribute to eDNA release that may contribute to gene transfer and subsequent environmental retention. Here, we investigated how different biocides affect the release of eDNA from mature biofilms of two opportunistic model strains *Pseudomonas aeruginosa* ATCC 27853 (PA) and *Staphylococcus aureus* ATCC 25923 (SA) and contribute to the hospital resistome in the form of surface and water contaminants and dust particles. The effect of four groups of biocides, alcohols, hydrogen peroxide, quaternary ammonium compounds, and the polymeric biocide polyhexamethylene guanidine hydrochloride (PHMG-Cl), was evaluated using PA and SA biofilms. Most biocides, except for PHMG-Cl and 70% ethanol, caused substantial eDNA release, and PHMG-Cl was found to block biofilm development when used at concentrations of 0.5% and 0.1%. This might be associated with the formation of DNA–PHMG-Cl complexes as PHMG-Cl is predicted to bind to AT base pairs by molecular docking assays. PHMG-Cl was found to bind high-molecular DNA and plasmid DNA and continued to inactivate DNA on surfaces even after 4 weeks. PHMG-Cl also effectively inactivated biofilm-associated antibiotic resistance gene eDNA released by a pan-drug-resistant *Klebsiella* strain, which demonstrates the potential of a polymeric biocide as a new surface-active agent to combat the spread of antibiotic resistance in hospital settings.

## 1. Introduction

The risk of multi-drug and pan-drug-resistant bacterial infections during or after a hospital stay is increasing worldwide. According to the WHO (2009), in general, health-care-associated infections in Europe lead to death in at least 2.7% of the cases and accounts for 135,000 deaths per year. According to the National Expert Group on Infection Control (https://negic.ua/eng/, accessed on 1 September 2021), there are up to 1 million cases of hospital-acquired infections and 50,000 deaths occurring in Ukraine each year. Each drug-resistant bacterium can contribute antibiotic resistance genes (ARGs), or resistome, to the hospital environment. This includes ARGs located within bacterial genomes (intracellular DNA) as well as cell-free ARGs in extracellular DNA (eARGs).

Microbial eDNA is ubiquitous and can be found everywhere where microbial life is present. It is released by the process of cell death and lysis and may be secreted by living cells, but is not enclosed in living cells, may not remain with the cells from which it originated, and may persist for some time before it is finally degraded [1]. eDNA is involved in microbial survival, often contributes to the structure of biofilms as a matrix component [2,3,4,5,6,7,8], and acts to preserve a dynamic pool of genes via horizontal gene transfer (HGT) within the biofilm community. ARGs are an important source for naturally occurring HGT [9,10,11] via transduction and conjugation. In contrast, the spread of eARGs by transformation and outer-membrane-vesicle-associated transport seems to be underestimated [1,12].

Over the last few years, investigations of eDNA distributions and its role in various resistomes have increased considerably. However, research mainly focuses on the wastewater environmental resistome, while the source and fate of eARGs in other environments, including hospital-associated niches, remain largely overlooked. For instance, while PubMed Central (https://www.ncbi.nlm.nih.gov/pmc, accessed on 1 March 2021) contains over 7000 references for “eDNA + water” and “eDNA + hospital”, less than 2000 references can be found for “eDNA + antibiotic resistant gene” and “eDNA + antibiotic resistant gene + hospital.” Nevertheless, an analysis of eARGs in aquatic sediments and water can potentially provide insights into the role of these eARGs as part of the environmental antibiotic resistome. eARGs persist more easily in sediments, which makes them more available for transformation [13]. HGT of such clinically significant eARGs (containing, for example, extended-spectrum beta-lactamases and carbapenem-resistant genes) can then transform environmentally harmless bacteria into pathogens that may then pose a threat to human health. A good demonstration of the importance of HGT in the evolution of human pathogens can be found in water-borne *Vibrio cholera*, where the ability to take up eDNA (competence) is induced by chitin, which is abundant in aquatic habitats [14].

In contrast, little is known about the impact of eDNA on the hospital resistome [13]. Hospital sewage systems are the source of persistent and recombinant *bla*KPC plasmids, which cause hospital-acquired infections in patients [15]. Hospital wastewater sludge is a prime source of eARGs, with a higher abundance of eARGs than normal wastewater sludge, pharmaceutical factory waste, lake sediments, or swine manure [16]. Hospital materials [17,18,19]; furniture [20]; and small items, such as stethoscopes [21], mobile phones [22,23,24,25], and keyboards [26], are also known to be the source of hospital-associated infections, and even dust plays an important part in the deposition and transmission of ARGs [27] in the hospital environment. Ironically, poor or incomplete cleaning of surfaces with bacterial colonies or biofilms with biocides may release intracellular DNA, which might then persist as eDNA until it degrades or is removed by subsequent cleaning (a combination of mechanical and chemical cleaning is needed to remove dried DNA effectively from surfaces [28]). While assessments of the effectiveness of disinfectants on biofilms largely concentrate on cell survival [29], little is reported about the amount of DNA released, whether it is soluble, whether it remains associated with cell debris or complexes with compounds (including disinfectants), or its persistence in the environment post cleaning.

Bacterial-biofilm-forming communities producing eDNA containing ARGs as well as eARGs no longer associated with biofilms, therefore, pose a significant problem in modern health-care environments. ARGs can spread throughout a hospital econiche by plasmid transfer, in the genomes of bacteriophages and as naked linear DNA [9,10]. There are only a limited number of studies in which the susceptibility of bacterial communities and eDNA to different biocides have been evaluated, and HGT may still be possible when cell-damaging biocides have been deployed but have failed to kill all bacteria in a biofilm [30,31]. The aim of our research was to evaluate the effect of different biocides on eDNA in biofilms produced by model hospital opportunistic pathogens, with a specific focus on the use of the polymeric biocide polyhexamethylene guanidine hydrochloride (PHMG-Cl) as a potential DNA-deactivating biocide.

## 2. Results

### 2.1. Effects of Different Biocides on DNA Release from PA and SA Biofilms

To investigate how much eDNA could potentially be released from biofilm contamination likely to be found in hospital environments following treatment with biocides, we used dehydrated 5-day-old *Pseudomonas aeruginosa* ATCC 27853 (PA) and *Staphylococcus aureus* ATCC 25923 (SA) biofilms to mimic dried biofilm material likely to be found on a variety of surfaces [32]. The preparation of the dehydrated samples was likely to cause some cell damage and the release of intracellular DNA, adding to the eDNA that had accumulated during the growth of the biofilms before drying. We considered both the pre-existing eDNA and freshy released DNA to be eDNA in the context of our assays, as eDNA is released by both living and dead cells [1]. Any further contribution to eDNA levels by biocides could then be assessed by comparison with water-only controls (which lacked biocide treatment). The water-only treatment might release some DNA from cells that survived the preparation of the dehydrated samples through osmotic shock, but it is more likely that in this control, DNA is solubilized from cell debris and the original eDNA of the biofilm (we note that both PA and SA are isolated from fresh and drinking water, indicating that they are adapted to hypo-osmotic conditions [33,34]).

We tested these biofilms with a number of biocides used frequently in the National Children’s Specialized Hospital Okhmatdyt (Kiev, Ukraine), including hydrogen peroxide; ethanol; isopropanol-based Desmanol; 0.05% chlorhexidine; and commercially available mixtures of quaternary ammonium compounds, such as Maxisan, Arquades-plus, and Sanikon; plus the polymeric cationic biocide polyhexamethylene guanidine hydrochloride (PHMG-Cl), which we had previously used in the synthesis of a new thermally stable polymeric biocide polyhexamethylene guanidine 2-naphtalenesulfonate to modify polyamide 11 [35]. After 1 h of exposure of the biofilms to the biocide, we isolated eDNA from the biofilms and measured eDNA yields compared to a water-only control treatment (Figure 1). This demonstrated that eDNA release from biofilms (i.e., pre-existing eDNA or DNA released from biocide-damaged cells) differed depending on both the biofilm and the biocide.

We also visualized the released eDNA by gel electrophoresis, which confirmed that PHMG-Cl reduces eDNA yield from PA biofilms compared to the water-only control treatment. The eDNA was a mix of high-molecular-weight DNA, with substantial amounts remaining in the wells after electrophoresis, and highly degraded DNA. In contrast, PHMG-Cl, alcohols, quaternary ammonium compounds, and 3% hydrogen peroxide were able to reduce the eDNA yield from SA biofilms (Figure 2), in agreement with our direct measurement of eDNA concentrations. 

Surprisingly, the eDNA released from the dehydrated biofilms after their exposure to biofilms was largely undamaged, as assessed by PCR of 16S rDNA (Figure 3). However, PCR amplification of the PHMG-Cl samples failed, suggesting that, in addition to reducing eDNA yield, this biocide was complexing with the eDNA to prevent the amplification of the target sequence. 

### 2.2. Anti-Biofilm Activity of PHMG-Cl against PA and SA Biofilms

Given the effectiveness of PHMG-Cl in inactivating the eDNA released from dehydrated biofilms, we tested PHMG-Cl to see if it could also suppress or inhibit the growth of PA and SA biofilms in situ. We treated 24 h old biofilms with different concentrations of PHMG-Cl and then determined metabolic activity using MTT (Figure 4). In this assay, the lowest concentration of PHMG-Cl that was not significantly different from the EtOH-treated (dead) negative control biofilm but significantly lower than the water-treated (live) positive control was biocidal. For the PA biofilm, the biocidal concentration was 0.1%, and for the SA biofilm, it was 0.5% PHMG-Cl. Increasing concentrations of PHMG-Cl also reduced eDNA yield from biofilms (Figure 5). At 0.5% PHMG-Cl, the eDNA yield was reduced to less than a quarter of that from the water-treated positive PA control biofilm and to around one-third of that from the SA control biofilm.

The spatial distribution of eDNA in PA and SA biofilms treated with water only and those treated with 0.5% PHMG-Cl was also investigated by CLSM, with PI used to visualize eDNA and dead cells and SYBR Green used to visualize intracellular DNA in living cells with undamaged membranes [36] (Figure 6). As was expected, the eDNA could be visualized in 3-day-old PA and SA biofilms and in these water-only controls, the DNA was specifically eDNA that had accumulated during the development of the biofilm, as no biocide treatment was applied that might have damaged cells and caused the release of intracellular DNA. However, PHMG-Cl had different effects on the biofilms, which may reflect the differences between PA and SA biofilm structures and cell wall organization. In PA biofilms, PHMG-Cl caused substantial lysis of cells and the formation of large eDNA filaments. Conversely, in SA biofilms, PHMG-Cl caused substantial cell damage and a lot of the eDNA on the top surface of the biofilm was lost. In both biofilms, cell damage appeared to result in decompaction and an increase in biofilm thickness. We noted that the SYBR Green signal was enhanced after PHMG-Cl treatment, presumably because the damage to cell walls and membranes allowed greater penetration of the cell and binding intracellular DNA (SYBR Green has a relative MW of approx. 510, compared with 670 for PI, explaining why the PI signal did not seem to show a similar increase in strength if it could also access cells through damaged membranes).

### 2.3. Inactivation of DNA by PHMG-Cl

PHMG-Cl was the only biocide found to prevent the PCR amplification of 16S rDNA target sequences from eDNA samples (Figure 3). We, therefore, speculated as to how eDNA inactivation might have occurred, given that measurable quantities of eDNA could still be recovered from PHMG-Cl-treated biofilms. We first extracted eDNA from PA biofilm samples in a manner that avoided contamination by intracellular DNA and used an electrophoretic mobility assay to determine whether PHMG-Cl formed DNA complexes that might prevent PCR amplification. PA eDNA treated with 0.001–0.01% PHMG-Cl did not affect the intrinsic charge of DNA molecules, but concentrations of 0.05–0.5% PHMG-Cl effectively neutralized the DNA, which was then lost during gel electrophoresis (Figure 7). However, it is not clear from this assay whether PHMG-Cl binding was co-operative or not, but we assume that 0.05% PHMG-Cl was sufficiently saturating to bind all surface sites. Dialysis or deproteinization of the eDNA–PHMG-Cl reaction mixture had little impact on electrophoresis, suggesting that these complexes were stable and not dependent on the presence of DNA-binding proteins that may have co-isolated with the eDNA (the presence of eDNA was confirmed by NanoDrop spectrophotometry, demonstrating that DNA was not lost during processing). PHMG-Cl also complexed with covalently closed circular (CCC) and linear forms of pC1-L plasmid DNA, suggesting that this process may not be too sensitive to DNA structure. Treatment of CCC and linear pC1-L also prevented PCR amplification of the LIF gene target sequence (Figure 8). We also included a mixture of PHMG-Cl-treated pC1-L DNA with untreated pC1-L DNA in this PCR assay, which demonstrated that once complexed with DNA, PHMG-Cl could not bind new DNA (suggesting a low dissociation constant for the DNA–PHGM-Cl complex) or directly interact with the Taq polymerase to prevent PCR amplification. These assays confirmed that treatment with 0.05% PHMG-Cl effectively inactivated a range of DNA, including eDNA as well as CCC and linear plasmid DNA. 

### 2.4. Inactivation of eARG by PHMG-Cl

To confirm the direct inactivation of a model ARG in eDNA by PHMG-Cl, we extracted eDNA from biofilms of a pan-drug-resistant *Klebsiella pneumoniae* hospital isolate (UHI KP 1633). We confirmed that UHI KP 1633 was a pan-drug-resistant strain and was resistant to carbapenems and colistin and we used real-time PCR to amplify the carbapenemase (KPC) resistance gene from eDNA treated with PHMG-Cl (Figure 9). Real-time PCR was only able to detect KPC in the eDNA water-only control and samples treated with 0.001–0.005% PHMG-Cl but not in samples treated with higher concentrations of PHMG-Cl, in agreement with our earlier tests of PA eDNA and pC1-L plasmid DNA.

### 2.5. DNA-Inactivating Activity of PHMG-Cl Adsorbed onto Plastic Surfaces

PHMG-Cl is known to have strong absorptive properties, and slow-release from surfaces can provide long-lasting antibacterial defense [37,38]. We speculated whether the inactivation of DNA we had observed with PHMG-Cl added to dehydrated and growing biofilms and eDNA also occurred when the eDNA was allowed to dry onto plastic surfaces treated with PHMG-Cl. We treated 96-well polystyrene plates with PHMG-Cl and added PA eDNA and stored them at room temperature for up to 28 days before recovery of DNA and gel electrophoresis (Figure 10). Although the total amount of DNA recovered decreased with age, sufficient DNA remained after 28 days to demonstrate that surfaces treated with 0.5–1.0% PHMG-Cl could still inactivate eDNA and that 0.05–0.1% PHMG-Cl could also protect surfaces and inactivate eDNA for 4–8 days. 

### 2.6. DNA Docking Assay

We undertook molecular docking using the DNA oligonucleotide 1DNE to investigate the mechanism that might allow the formation of DNA–PHMG-Cl complexes. 1DNE has been previously used in several studies to investigate the interaction of various ligands with DNA [39,40,41]. As we were interested in identifying PHMG-Cl interactions in both AT- and CG-rich regions, we modeled PHMG-Cl as a dimer following the approach used in studying DNA–PHMB dimer interactions [42]. We obtained a 1DNE DNA–PHMG dimer complex that was stabilized by six hydrogen bonds (1.97–3.20 Ǻ) and five electrostatic bonds (1.97–3.20 Ǻ), with five additional longer-range electrostatic interactions (4.42–5.13 Ǻ) occurring via the PHMG amino groups (Figure 11). The DA7, DT6, DT8, DC9, DT18, DA19, and DC21 nucleotide bases in 1DNE play a key role in stabilizing the complex. Our docking results showed the formation of the DNA–PHMG dimer complex with estimated binding energies of 7.8 kcal/mol predominantly at the AT base pair region of 1DNE. 

We validated our approach by redocking the DNA intercalator Netropsin after ligand randomization [39,43] to 1DNA with an estimated binding energy of −7.7 kcal/mol after ligand randomization with an RMSD value for all atoms of 0.2 Ǻ. The similarity between the two estimated binding energies suggests that we selected the correct docking strategy for the PHMG dimer.

## 3. Discussion

### 3.1. Impact of Biocides on eDNA 

Biofilms are a form of bacterial colonization of the environment [44,45,46,47]. eDNA is an extracellular matrix polymer found in the biofilms of many important opportunists, such as *Acinetobacter baumannii*, *Enterococcus faecalis*, *Helicobacter pylori*, *P. aeruginosa*, and *Staphylococcus* spp. [2,3,4,5,6,7,8]. It is an important structural element and is stabilized by functional amyloids and cross-linked with polysaccharides [8,48,49]. eDNA is also a source of nutrients and provides antibacterial, antibiotic-resistant, and regulatory activities [50,51,52] and has good adhesive properties in *P. aeruginosa* and *Bacillus cereus* biofilms in particular [4,53]. It has been traditionally considered as an inevitable attribute of microbial contamination, and in recent years, more attention has been paid to eDNA as a source of ARGs contributing to hospital resistomes. eDNA has finally been recognized as a contaminant that should be controlled to reduce HGT intensity and resistomes [7,54,55]. Similar to other resistance factors, eDNA in biofilms can be enhanced by sublethal doses of antimicrobials [56,57,58] and biocides [9,59], which result in hospital contamination [16]. It is, therefore, likely that the use of some biocides in hospitals may be counter-productive in controlling biofilms, eDNA, HGT, and resistomes. 

A better understanding of how biocides affect the release of eDNA from biofilms and contribute to surface and water contaminants and dust particles is required. In this research, the effect of four groups of biocides on eDNA release from dehydrated biofilms produced by two model hospital pathogens, *Pseudomonas aeruginosa* ATCC 27853 and *Staphylococcus aureus* ATCC 25923, was studied. Interestingly, almost all the biocides tested, except for PHMG-Cl and 70% ethanol, caused substantial eDNA release that was a mixture of pre-existing eDNA accumulated during the growth of the biofilm samples and DNA freshly released from damaged cells as the result of the application of the biocide. The amount of eDNA released from these biofilms might be explained by structural differences between biofilms. PA biofilms are rich in eDNA, which plays not only a structural role but also an important physiological role, providing phenotypic resistance [48,50,51]. 

### 3.2. PHMG Blocks Biofilm Development

Cationic polymers are now being considered as a new generation of biocides due to their enhanced antimicrobial activity, as well as their low toxicity to human cells, compared to common low-molecular cationic surfactants [60,61]. A wide variety of biocidal cationic polymers comprising quaternary ammonium, pyridinium, phosphonium, and guanidinium cations in main chains or as pendant groups has been reported [60,61,62,63]. The high activity of cationic polymers against microorganisms is caused by the presence of multiple positive charges within a single molecule that are able to compensate the negative charges present on the outer cell membranes of microbes. Due to these strong electrostatic interactions, polycations are able to attack the cellular envelope and subsequently associate with the head groups of the acidic phospholipids. The presence of hydrophobic aliphatic chains in the cationic polymer structure ensures a better partition to the hydrophobic regions of the phospholipids membrane, resulting in a change in membrane permeability and lethal leakage of cytoplasmic materials [60,64]. So far, this has been the only established way of the action of polycationic antimicrobial polymers confirmed by numerous studies [60,65]. 

Polymeric cationic biocides incorporating guanidinium cations in the polymer backbone, such as polyhexamethylene biguanide hydrochloride (PHMB-Cl) and polyhexamethylene guanidine hydrochloride (PHMG-Cl), are often considered as cost-effective alternatives to common inorganic antimicrobial agents as they have high efficacy in killing antibiotic-resistant bacteria and fungi and low cytotoxicity [66,67,68,69,70]. Guanidinium-based cationic polymers are widely used as effective disinfectants in cooling systems, swimming pools, and hospitals; in personal hygiene products; as well as in the food industry [66,68,70,71].

Although bacterial-biofilm-forming communities pose a significant problem in modern healthcare environments, there are a limited number of studies investigating their susceptibility to existing biocides and cationic polymers such as PHMG-Cl have been tested on *S. aureus* in liquid cultures (planktonic state) as well as in biofilms [72] and have demonstrated an anti-biofilm activity that is significantly higher compared to that of common antimicrobial agents, including benzalkonium chloride, cetrimide, chlorhexidine, and cetylpyridinium chloride [73]. PHMG-Cl also demonstrated bactericidal advantages over chlorhexidine digluconate against ESKAPE bacteria [74].

To evaluate the biocidal effectiveness of PHMG-Cl, we measured the total metabolic activity of PA and SA biofilms, which confirmed that it is an effective antibacterial compound. Our testing again demonstrated a difference between PA and SA biofilms, with SA biofilms more resistant to PHMG-Cl, with a minimal effective anti-biofilm concentration of 0.5%, which was 5× lower than that needed for PA biofilms. However, the mechanism behind the effectivity of PHMG-Cl against biofilms or biofilm-resident cells is not yet understood for either Gram-negative or Gram-positive opportunistic pathogens. We noted that PHMG-Cl could effectively release pre-existing eDNA and DNA from damaged cells from the dehydrated biofilms that we used as a model surface contamination as well as from fully hydrated and growing biofilms, suggesting that this biocide might work well with both developing biofilms as well as biofilm residue. 

### 3.3. PHMG Binds to DNA and Inhibits Its Functionality 

Our molecular docking analysis of a PHMG dimer with a model DNA oligonucleotide suggests that the high anti-biofilm activity of PHMG-Cl may be associated with specific DNA binding in high-AT regions. DNA binding is also the likely explanation of why PHMG-Cl-treated eDNA and pC1-L plasmid did not enter agarose gels during electrophoresis and the failure of PCR to amplify target sequences. PHMG-Cl was able to inactivate genes, including ARGs, when used at 0.01–0.05%, and plastic surfaces treated with PHMG-Cl continued to show DNA deactivation for up to 28 days after application. For high-molecular-weight and degraded eDNA, there was a direct correlation between eDNA and PHMG-Cl concentrations needed to inactivate the DNA and 0.01% PHMG-Cl also inactivated CCC and linear plasmid DNA. A similar DNA-inactivation effect was observed earlier for polyamide films containing 7–10% PHMG-NS [32], where only the PHMG at the surface of the film was actively in contact with biofilm eDNA. There are some reports that polyhexamethylene biguanidine (PHMB) might bind selectively and condense bacterial chromosomes [42,74]. 

Selective binding and condensation of intracellular DNA might be one more mechanism inhibiting bacterial growth [74]. To investigate this effect and the spatial distribution of both eDNA and intracellular DNA in PA and SA biofilms, we applied the classical live/dead staining routinely used in microbiology [36], with PI used to visualize eDNA and the DNA of cells with damaged membranes and the membrane permeable SYBR Green to stain all DNA. As both dyes bind in the minor groove of DNA, fluorescence resonance energy transfer (FRET) results in eDNA and dead cells fluorescing only with a red emission [75]. We observed eDNA in both PA and SA biofilms as expected and found that PHMG-Cl treatment differentially impacted eDNA and biofilm structure. PHMG-Cl caused cell lysis and the formation of large eDNA filaments, and biofilm thickness increased in areas of acute cell damage. In SA biofilms, it appeared that a lot of the eDNA from the top surface of the biofilm had been lost despite the gentle washing we used in the treatment of the samples. Finally, we noted that the effects of PHMG were not confined to eDNA only. Similar to others [76], we noticed microscopic evidence of cell wall damage after PHMG-Cl treatment (Figure 12), confirming the generally accepted model of PHMG-Cl activity in which cell walls and membranes are affected. This is confirmed by our observation that after PHMG-Cl treatment, the SYBR Green signal was enhanced because of the easier access to the intracellular DNA through damaged membranes.

## 4. Materials and Methods

### 4.1. Synthesis of the Polymeric Biocide Polyhexamethylene Guanidine Hydrochloride (PHMG-Cl)

A mixture of guanidine hydrochloride (20 g, 0.21 mol) and hexamethylenediamine (23.1 g, 0.2 mol) was placed in a round-bottomed flask (500 mL) equipped with a mechanical stirrer (Figure 1). This mixture was heated to 100 ºC and the melt was stirred for 4 h at this temperature. Further, the reaction was carried out for 4 h at 130–140 ºC, and finally 4 h at 180 ºC to obtain a highly viscous melt of PHMG-Cl. A vitreous solid was obtained after cooling the mixture to room temperature, which was then dissolved in water (200 mL), filtered, and precipitated by the addition of a saturated aqueous solution of sodium chloride (100 mL). PHMG-Cl was isolated by the decantation of the water solution and dried at 140 °C for 24 h before being powdered in a porcelain mortar. The final product had a melting point of 134–136 ºC and an intrinsic viscosity of 0.09 dL/g in 0.1 N NaCl at 25 °C. The molecular weight of PHMC-Cl was calculated using the Mark–Houwink equation [η] = K × M^α^, where [η] is intrinsic viscosity, M is the viscosity-average molecular weight, α and K are parameters whose values depend on the nature of the polymer and the solvent. For the PHMG-Cl–water system, K = 1.83 × 10^–3^ and α = 0.38 at 25 °C [77]. Thus, the viscosity-average molecular weight of the synthesized PHMG-Cl was found to be 28,000. 

NMR spectra of PHMG-Cl were recorded in DMSO-d6 on a Varian Gemini-2000 (400 MHz) spectrometer with the following results: ^1^H NMR (400 MHz, DMSO-D6): δ = 1.32 (m, 4H, CH_2_), 1.47 (m, 4H, N-CH_2_CH_2_), 3.16 (m, 4H, N-CH_2_), and 7.1–8.1 (br s, 4H, NH); ^13^C NMR: 25.3 (CH_2_), 28.1 (N-CH_2_CH_2_), 40.5 (N-CH_2_), and 156.6 (NH).

### 4.2. Biocides Used in the Assays

Seven biocides were used in addition to PHMG-Cl, i.e., 35% hydrogen peroxide (Khimpostachannia, Kharkiv, Ukraine); 96% ethanol (UkrSpirt Trade, Kiev, Ukraine); 0.05% chlorhexidine (Zdorovie, Kharkiv, Ukraine); Maxisan (Interdez, Kiev, Ukraine), which contained not less than 50% of a mix of four quaternary ammonium compounds; Arquades-plus (O. L. KAR, Vinnitsa, Ukraine), containing 10% dimethyldialkylammonium chloride, 5% didecyldimethylammonium chloride, and 2.5% tetrasodium salt); Desmanol (Schülke & Mayr GmbH, Germany), containing 75% isopropanol; and Sanikon (Interdez, Kiev, Ukraine), containing not less than 5.5% of a mix of four quaternary ammonium compounds. 

### 4.3. Microorganisms and Culturing Conditions

Two model hospital opportunistic pathogens, *Pseudomonas aeruginosa* ATCC 27853 (PA) and *Staphylococcus aureus* ATCC 25923 (SA), were used to study the effects of biocides on established biofilms and eDNA yield. The Ukrainian hospital isolate *Klebsiella pneumoniae* 1633 (KP) was recovered from a patient and identified as a pan-drug-resistant (PDR) strain using antibiotic disc diffusion assays and EUCAST 2021 v.11.0 breakpoints. An AST-N332 card was used to confirm the PDR strain phenotype with the VITEK 2 Advanced Expert System. Antimicrobial susceptibility testing of CMS was performed by a SensiTest Colistin (Liofilchem, Roseto degli Abruzzi, Italy) broth microdilution assay and the results interpreted according to EUCAST breakpoints. Bacterial strains were cultured aerobically at 37 °C with shaking in a Luria–Bertani (LB) medium to provide inoculum [78]. Preliminary biofilm experiments showed that the surface–liquid interface biofilms produced by KP, PA, and SA were readily dislodged from glass vials or 96-well plates (this is an acknowledged problem in crystal staining biofilms) [79]. We therefore decided to produce biofilm samples for the assay without rinsing or washing. We noted that the samples will therefore contain a proportion of planktonic cells but that this will have a limited impact on our assays, as KP and SA lack flagella and the initial inoculum will not remain in the liquid column for long and all subsequent growth must be in biofilms. While PA is capable of flagella-mediated swimming, it is highly aerotaxic and most cells will accumulate at the air–liquid interface and in the meniscus region where biofilm growth dominates. We used water-only treatments to provide positive controls for some assays but do not believe these brief exposures to distilled water would lead to substantial cell lysis through hypo-osmotic shock, as both PA and SA are often isolated from fresh and drinking water and must, therefore, be able to cope with this stress [33,34]. For the biofilm inactivation assay, LB cultures were diluted to 10% and 200 µL aliquots transferred to the wells of a 96-well plate that was incubated at 37 °C for 24 h before the biocide inactivation assay. For CLSM and the eDNA yield assay, 10% dilutions were used to inoculate 30 mL glass vials (microcosms) containing 2–5 mL LB that were incubated statically at 37 °C for 3–5 days to produce biofilms [80]. The biofilms were air-dried at room temperature for 14 days before eDNA yield analysis. For CLSM, biofilms were grown in 5 mL stationary glass vials and samples recovered using a pipetter with cut 200 µL tips. The biofilm samples were gently placed onto microscope glass slides and the liquid drained off using filter paper. They were then treated with a PHMG-Cl solution for 30 min and gently washed with water using a pipetter to remove the biocide but limit any further physical damage before staining and imaging (see below). 

### 4.4. Plasmids Used in the Study

The pC1-L plasmid containing the human LIF gene [81] was used to investigate the effect of biocides on covalently closed circular and linear plasmid DNA.

### 4.5. Biofilm Metabolic Assay

Aliquots of an aqueous PHMG-Cl solution were added to the wells of a 96-well plate with 24 h PA and SA biofilm cultures at final concentrations of 5%, 1%, 0.5%, 0.1%, 0.05%, 0.01%, 0.005%, and 0.001%, with eight replicates per treatment. These were incubated for 1 h at room temperature. A negative control sample was produced using 50% ethanol and a positive (live) control produced by adding sterile distilled water. MTT solution (Sigma-Aldrich, UK) was then added to each well to a final concentration of 0.05% and the mixture incubated at 37 °C for 3 h. Biofilms were removed from each well and placed in 1.5 mL plastic tubes, which were then centrifuged at 13,000× *g* for 15 min in an Eppendorf 5424 microcentrifuge (Eppendorf, Germany). The supernatant was discarded, and the pellet dissolved in 500 µL of DMSO. Metabolic activity was evaluated using absorbance measurements at 570 nm in a BioTek ELx800 microplate spectrophotometer (BioTek Instruments, Inc., Winooski, VT, USA). Net biofilm metabolic activity was calculated by subtracting the negative control values.

### 4.6. eDNA Yield Assay

Biocides were tested at different concentrations to determine eDNA yields. For this, 1 mL of 0.5%, 0.1%, and 0.05% PHMG-Cl; 6% and 3% hydrogen peroxide; 70% ethanol; 0.05% chlorhexidine; 0.25% Maxisan; 0.5% Arquadez-plus; 0.5% Desmanol; and 0.5% Sanikon were added to dehydrated PA and SA biofilms at room temperature and the mixtures kept for 1 h, with three replicates per treatment. A water-only treatment was again used as the positive control. eDNA was extracted from the liquid phase of each sample. Proteins were first removed with an equal volume of chloroform and DNA precipitated with 70% ethanol and 0.3 M sodium acetate (pH 8.0). DNA was then dissolved in TE buffer. The nucleic acid concentration was measured by a NanoDrop 2000 spectrophotometer (Thermo Fisher Scientific, Inc., Wilmington, DE, USA). DNA samples were then visualized by 1.2% agarose-TA gel electrophoresis and EtBr staining. GeneRuler 1 kb DNA Ladder (Thermo Fisher Scientific, Vilnius, Lithuania) was used as a size marker.

### 4.7. Biofilm eDNA Isolation 

PA and KP biofilms were grown in 5 mL LB static vials at 37 °C for 5 days. The biofilms were then disintegrated by agitation for 5 min. Then, 5 mL of the biofilm suspension was centrifuged at 13,000× *g* for 15 min using a microcentrifuge and the supernatant recovered. The eDNA in the supernatant was precipitated with 70% ethanol and 0.3 M sodium acetate (pH 8.0) and then dissolved in TE buffer. The nucleic acid concentration was measured by NanoDrop.

### 4.8. PHMG-Cl Effects on eDNA and pC1-L Plasmid DNA 

The eDNA isolated from PA 27853 and pC1-L DNA was treated with different PHMG-Cl concentrations. For this, 7 µL (6.7 µg) of eDNA and 7 µL (2.8 µg) of linear or covalently closed circular (CCC) pC1-L DNA was mixed with 3 µL of DNA loading buffer (Thermo Fisher Scientific, Vilnius, Lithuania) and final concentrations of 0.01% and 0.05% PHMG-Cl. Samples were placed in D-0530 dialysis tubing (Sigma, Saint Louis, MO, USA) and the samples dialyzed together to reduce the PHMG-Cl concentration by 10–7× in TE buffer. The DNA samples were visualized by gel electrophoresis.

### 4.9. PCR of 16S rDNA, LIF, and KPC Sequences

PCR was used to determine the effect of biocide treatment on DNA by amplifying 16S rDNA, LIF, and *Klebsiella pneumoniae* carbapenemase (KPC) target sequences. To amplify 16S sequences from eDNA samples, 25 µL PCR reaction mixtures (Taq PCR Kit, New England Biolabs, Ispwich, MA, USA) with 27F (5′-AGAGTTTGATCMTGGCTCAG-3′) and 1492R (5′-TACGGYTACCTTGTTACGACTT-3′) primers were used [82]. Forward (5′-ATGAAGGTCTTGGCGGCAGG-3′) and reverse (5′-ACCTCCTGCTAGAAGGCCTG-3′) primers were used to amplify the LIF gene from pC1-L samples. PCR conditions involved an initial stage of 5 min at 95 °C. This was followed by 30 cycles at 95 °C for 30 s, 57 °C for 30 s, and 72 °C for 30 s and a final stage at 72 °C for 5 min. Purified genomic DNA was used as a positive control and the PCR products visualized by gel electrophoresis. KPC sequences were amplified using the AmpliSense MDR KPC/OXA-48-FL reagent kit (AmpliSense, Moscow, Russia) and the CFX96 Touch Real-Time PCR Detection System (Bio-Rad, Hercules, CA, USA). 

### 4.10. Effect of PHMG-Cl-Treated Plastic Surface on eDNA

First, 10 µL of 1%, 0.5%, 0.1%, and 0.05% PHMG-Cl was placed into the wells of 96-well polystyrene plates, with three replicates per treatment. Then, the plates were allowed to dry at room temperature and left for 4, 8, 11, 15, 20, 25, and 28 days under the same conditions. Finally, 10 µL of 880 ng/µL eDNA from PA biofilms was added to the wells and the mixture incubated at room temperature for 1 h before nucleic acid concentrations were measured by NanoDrop and visualized by gel electrophoresis. 

### 4.11. Molecular Docking Assay

Molecular docking was performed using the model DNA dodecamer (CGCGATATCGCG) used to study DNA-binding compounds (RCSB Protein Data Bank 1DNE: https://www.rcsb.org/structure/1dne, accessed on 1 March 2021) [83]. Chains A and B were used for docking, and the water molecules and the ligand were removed from the crystal structure using Accelrys DS 4.0 [84]. AutoDock Tools (ADT) 1.5.6 [85] was used to make the PHMG dimer ligand and add polar hydrogens to the DNA. The noBondOrder method was used to renumber all atoms including the new hydrogen atoms, and the Gasteiger method was used to calculate charges. ChemAxon Marvin Sketch 5.3.735 [86] was used to optimize the PHMG dimer structure. Energy minimization and optimization of the PHMG dimer ligand were performed by MOPAC2016 [87] using the Auto Optimization Tool (MMFF94s force field) [88]. Partial charges and torsion angles of the ligand were changed using ADT. The DNA structure and the PHMG dimer were used for molecular docking by AutoDock Vina 1.1.2 [89]. A grid box of 30 × 30 × 30 points was used with a spacing of 1 Å. The analysis and visualization of interactions were performed by Accelrys DS.

### 4.12. Confocal Laser Scanning Microscopy (CLSM)

Biofilm samples were stained with 5 µL of 100× SYBR Green (Invitrogen, Waltham, MA, USA) and 1 mM propidium iodide (PI) (Sigma, Gillingham, UK). No additional washing was applied so as to limit the physical disruption of biofilm structures through liquid movement. The samples were not fixed, and a cover slip was placed over the stained samples before imaging. CLSM analysis was undertaken using a Leica TCS SPE Confocal system with a coded DMi8 inverted microscope (Leica, Mannheim, Germany) and Leica Application Suite X (LAS X) Version 3.4.1. Images were acquired using excitation at 488 nm and emission collected at 490–580 nm for SYBR Green and excitation at 532 nm and emission collected at 537–670 for PI.

### 4.13. Transmission Electron Microscopy (TEM)

An overnight PA culture was diluted to 20% and PHMG-Cl added to final concentrations of 5%, 1%, 0.5%, 0.1%, 0.05%, and 0.01%. Samples were incubated at room temperature for 20 min before a 10 µL aliquot was placed onto a formvar covered grid and dried at room temperature. Then, 10 µL of 1% uranyl acetate (Sigma-Aldrich, Gillingham, UK) was dropped onto each grid and dried with filter paper. TEM was performed with a JEM-1400 transmission electron microscope (JEOL, Tokyo, Japan).

### 4.14. Statistical Analysis

Replicate data were processed using the statistical software package OriginPro 7.0 and MS Excel for Windows. All results are presented as the mean ± standard deviation. A value of *p* < 0.05 was considered statistically significant.

## 5. Conclusions

Our investigations clearly demonstrate that the biocides commonly used in hospitals can have a significant impact on the release of eDNA from bacterial biofilms. Some biocides, in particular PHMG-Cl, were found to block biofilm development and were able to complex with DNA in a manner predicted by molecular docking assays. PHMG-Cl binding to DNA altered the electrophoretic mobility of both high-molecular-weight and plasmid DNA and prevented the amplification of a target ARG gene from the eDNA isolated from a Klebsiella biofilm. PHMG-Cl was also found to inactivate DNA when used to treat plastic surfaces, up to 28 days after application. These findings demonstrate the potential of PHMG-Cl as a surface-active agent that can be used in hospital settings to help reduce the spread of antibiotic resistance by inactivating the eDNA commonly found in bacterial biofilms and by limiting the development of biofilms themselves.

## Data Availability

All data have been included in this manuscript (figures).

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
