# Peer review of "eDNA Inactivation and Biofilm Inhibition by the PolymericBiocide Polyhexamethylene Guanidine Hydrochloride (PHMG-Cl)"

_ijms, 2022, doi:10.3390/ijms23020731_

Round 1

Reviewer 1 Report

MS: ijms-1486257

"eDNA inactivation and biofilm inhibition by the polymeric bi-2 ocide polyhexamethylene guanidine hydrochloride (PHMG-Cl)"

The possibility for horizontal gene transfer within biofilms has long been recognised as risk for the spread of antibiotic resistance genes. Such spread is enhanced by the space closeness of bacterial cells and might be performed via extracellular DNA among other mechanisms. This justifies the focus of the MS on the possibility to select appropriate disinfectants capable to destroy eDNA in the biofilm matrix. The biocide PHMG-Cl is the main focus of the study, and is compared with other biocides.

The complicated structure of bacterial biofilms creates difficulties in both reproducibility of results, and comparability between results from different laboratories. Therefore detailed description of methodologies has been put forward as a requirement for biofilm publications https://doi.org/10.1016/j.bioflm.2019.100010 .

In the study under review, this is a serious shortcome  - the methodology description of the separate experiments is often obscure and not given in clear detail. The way in which the experimental approach is described often creates difficulties for the unequivocal interpretation of the results due to the possible interference of factors that were not taken into account when planning the controls of the experiments.

There are questions regarding the biofilm cultivation. Two approaches for biofilm cultivation are mentioned in M&M: on 96-well microtitre plates, and using "microcosm". The latter term is somewhat confusing, and the authors should specify the use of it for the purposes of their study. A common use in literature of the term "microcosm" is for nature-isolated microbial communities, most often dental, applied in biofilm and other experiments. In the present MS the authors analyze single-species biofilms - cultivated in microcosm?! I consulted previous publications of the authors of this MS, it seems that this term has been used for more likely for either closed vials, or biofilm cultivated in closed vials, right? To make the methodology description more clear, the use of this term should be specified in M&M, together with description of the individual steps. Of importance, how did the biofilm develop under these "microcosm" conditions - on the solid-liquid or, more likely, on the air-liquid interface (pelicle)? This could make difference with the solid-liquid-interface biofilm developed in the 96-well plates.

In the DNA yield assay,when air-dried biofilm was used, it is not clear what is the amount of the air-dried biofilm that was used for evaluation and comparison between the effects on the different biocides on the eDNA from the dry biofilm biomass. During treatments of air-dried biofilms with the tested biocides, it is quite likely that some of the disinfectants or treatment procedures could alter bacterial cell integrity and therefore, the contribution (or lack of contribution) of intracellular iDNA to the pooled eDNA samples should not be overlooked, and should be monitored and supported by appropriate controls. One such control should be eDNA from otherwise untreated fresh hydrated mature biofilms in parallel with the "water-treated only" control.

The 96-well cultivation has been applied in this study for testing the viability of biofilm cells. The usual practice when testing biofilms grown in this way is to remove the unattached "plankton" cells from the wells in order to be sure you examine the biofilm. From the description of the experiment in M&M it appears that the authors added the biocide to the wells containing the liquid medium, and did not remove the unattached, "plankton" cells from the wells, and also did not wash the wells prior to the test, to make sure they are examining the biofilm. The MTT reagent was added also like this. The way the methodology is written it appears that the results are for free-swimming bacteria as well, and not the biofilm.

Author Response

We appreciate the fact that this Reviewer feels that our work is justified and is largely focussed on the activity of PHMG. We acknowledge that producing and assaying the effects on biofilms is difficult, but in our work, as in most other research publications, we have made use of experimental replicates to reduce variation between biofilm structure / biomass / cell condition at the start of each experiment and have included the appropriate controls in each case. We note with interest the Allkja et al. (2020) reference provided.

  • In the study under review, this is a serious shortcome  - the methodology description of the separate experiments is often obscure and not given in clear detail. The way in which the experimental approach is described often creates difficulties for the unequivocal interpretation of the results due to the possible interference of factors that were not taken into account when planning the controls of the experiments.

Response   We are disappointed that this Reviewer feels that some of our descriptions of methodology are obscure or not clear but note that the other Reviwers did not raise such concerns. Our biofilm assays describe starting inoculum, incubation conditions and time, replicates, controls and end-assays, and the Results section provide further details of experimental context, results, and statistical analyses (very much in line with Allkja et al. 2020).

  • There are questions regarding the biofilm cultivation. Two approaches for biofilm cultivation are mentioned in M&M: on 96-well microtitre plates, and using "microcosm". The latter term is somewhat confusing, and the authors should specify the use of it for the purposes of their study. A common use in literature of the term "microcosm" is for nature-isolated microbial communities, most often dental, applied in biofilm and other experiments. In the present MS the authors analyze single-species biofilms - cultivated in microcosm?! I consulted previous publications of the authors of this MS, it seems that this term has been used for more likely for either closed vials, or biofilm cultivated in closed vials, right? To make the methodology description more clear, the use of this term should be specified in M&M, together with description of the individual steps. Of importance, how did the biofilm develop under these "microcosm" conditions - on the solid-liquid or, more likely, on the air-liquid interface (pelicle)? This could make difference with the solid-liquid-interface biofilm developed in the 96-well plates.

Response   We apologise for the confusion we have caused here using the term ‘microcosm’. In our previous research, we have used ‘microcosm’ as other groups have done to refer to glass vials used to produce biofilms – both surface-liquid interface biofilms investigated here, as well as air-liquid interface biofilms we have focussed on in research using pseudomonads and soil-wash based bacterial communities. We had referred to ‘microcosms’ twice in our original submission. In order to avoid confusion, we have replaced the first with ‘glass vials (microcosms)’ and the second simply with ‘vials’.

  • In the DNA yield assay,when air-dried biofilm was used, it is not clear what is the amount of the air-dried biofilm that was used for evaluation and comparison between the effects on the different biocides on the eDNA from the dry biofilm biomass.

Response   In our DNA yield assays, we did not weigh individual biofilms produced in glass vials or 96-well plate, as they are simply too fragile to remove and too light to weigh individually. Instead, in all our biofilm-based assays, we pre-grew biofilms in replicate before treatment (including controls), and where appropriate, used experimental replicates of the pre-grown biofilms to account for any variation in biofilm structure / biomass / cell condition that occurred before any treatment. All controls are indicated in the Figure legends and in the Discussion where appropriate.

  • During treatments of air-dried biofilms with the tested biocides, it is quite likely that some of the disinfectants or treatment procedures could alter bacterial cell integrity and therefore, the contribution (or lack of contribution) of intracellular iDNA to the pooled eDNA samples should not be overlooked, and should be monitored and supported by appropriate controls. One such control should be eDNA from otherwise untreated fresh hydrated mature biofilms in parallel with the "water-treated only" control.

Response   We apologise for the confusion we have caused here and make the point that the eDNA we are analysing is the direct result of biocide application – we make this clear in the opening sentence of the Results by asking whether eDNA could be released from biofilms following treatment with biocides (Lines 99-100). Our comparison of water-only treatment and biocide-treated biofilms clearly shows that biocides damage biofilm structure and /or cell structure to release more DNA (by definition, becoming eDNA). We have added a comment in parentheses to make this point clear.

We do not feel that an additional negative control using ‘fresh’ biofilms as suggested is needed, as in our experiments we are comparing eDNA release from our experimental treatments with the water-only control. This is important, as our dehydrated biofilms are a model for dried biofilm contamination found in hospitals rather than actively growing biofilms.

  • The 96-well cultivation has been applied in this study for testing the viability of biofilm cells. The usual practice when testing biofilms grown in this way is to remove the unattached "plankton" cells from the wells in order to be sure you examine the biofilm. From the description of the experiment in M&M it appears that the authors added the biocide to the wells containing the liquid medium, and did not remove the unattached, "plankton" cells from the wells, and also did not wash the wells prior to the test, to make sure they are examining the biofilm. The MTT reagent was added also like this. The way the methodology is written it appears that the results are for free-swimming bacteria as well, and not the biofilm.

Response   We acknowledge the concerns raised here by this Reviewer regarding the potential problems raised by mixed planktonic and biofilm-based cell populations in 96-well–based assays. We chose not to wash our pre-grown biofilms before biocide treatment as the biofilms we were examining were poorly attached and were easily displaced even with gentle washing and agitation (this issue has been acknowledged in biofilm assay reviews, e.g., Azeredo et al. Critical review on biofilm methods. Crit. Rev. Microbiol. 2017; 43:313-351). However, the biofilms we have analysed are unlikely to be significantly affected by the presence of substantial numbers of ‘planktonic’ cells. P. aeruginosa cells are capable of flagella-based motility and will move up an oxygen gradient and attach to vial walls at the meniscus, where they develop surface-liquid interface biofilms with some (weak and very glutinous) extension across the liquid surface. As cultures mature, the oxygen gradient becomes more extreme, and more cells are found in the biofilm compared to the liquid column (as demonstrated by us for P. fluorescens, another biofilm-forming pseudomonad). In contrast, S. aureus and K. pneumoniae do not have flagella and therefore cannot remain in a static liquid column (i.e., technically they cannot produce ‘free swimming’ planktonic cells). For these two strains, apart from the initial inoculum at the start of the incubation, all cells found on the vial walls or wells will be incorporated into a biofilm at the end of the experiment.

We have modified the Methods to make this clear and added an explanatory section.

All the changes marked in red.

Reviewer 2 Report

the paper eDNA inactivation and biofilm inhibition by the polymeric biocide polyhexamethylene guanidine hydrochloride (PHMG-Cl), prepared by Olena V. Moshynets et al., presents novel and interesting results that deserve to be published after minor improvements:

  1. figure 1 - please increase the font size of the text. is very hard to be read
  2. figure 6 - please add a higher scale bar, it is very hard to read the actual one.

Author Response

We appreciate that this Reviewer finds our work novel and interesting.

  • Figure 1 - please increase the font size of the text. is very hard to be read.

Response   This has been done as requested.

  • Figure 6 - please add a higher scale bar, it is very hard to read the actual one.

Response   This has been done as requested.

Reviewer 3 Report

The present manuscript describes a study of the role of different biocides on the eDNA release from S. aureus and P.aeruginosa biofilms and their deactivation of this eDNA. Further, the role of the polymeric biocide PHMG-Cl on the deactivation of eDNA was investigated in more detail also with respect to a molecular simulation on the binding energy was investigated. The topic of this study is very interesting and an important addition to the existing knowledge on the role of biocides particularly in hospitals. The work is scientifically sound and should be published after addressing a few issues.

  1. PHMG-Cl is the central molecule in this study. It would be helpful to have some information of the molecular weight of this polymer, because this is particularly important, when discussing polymer/DNA complexes. The higher the molecular weight of the polymer, the more stable is the complex (multi-binding).
  2. The authors state on page 11 that cationic polymers are used as disinfectants. It would enhance the quality of the discussion, if not only PHMG is cited here, but also other, particularly hydrophilic polycations (Strassburg et al Macromolecular Bioscience 15 (12), 1710-1723 (2015)), which are also available in biodegradable form (Krumm et al ACS Applied Materials and Interfaces 12 (19) 21201–21209 (2020)).
  3. Further, the discussion on page 12 suggests that polycations disrupt cell walls and membranes. This should be discussed in more detail, particularly, because hydrophilic polycations are not disrupting cell membranes, but rather cross them without causing damage. The damages found in electron microscopy on dried samples are usually either artefacts or damages induced by the cells themselves indirectly caused by the biocidal action of the polymers, e.g., cell wall damage induced by inhibition of the teichoic acids for Gram-positive bacteria.

Author Response

We appreciate the fact that this Reviewer finds our work very interesting and that it is scientifically sound.

  • PHMG-Cl is the central molecule in this study. It would be helpful to have some information of the molecular weight of this polymer, because this is particularly important, when discussing polymer/DNA complexes. The higher the molecular weight of the polymer, the more stable is the complex (multi-binding).

Response   We have calculated the MW of PHMG-Cl and this is now reported in the Methods.

  • The authors state on page 11 that cationic polymers are used as disinfectants. It would enhance the quality of the discussion, if not only PHMG is cited here, but also other, particularly hydrophilic polycations (Strassburg et al Macromolecular Bioscience 15 (12), 1710-1723 (2015)), which are also available in biodegradable form (Krumm et al ACS Applied Materials and Interfaces 12 (19) 21201–21209 (2020)).

Response   Thank you for these suggestions. We have added a new paragraph to the Discussion and have included these and several other new references in support.

  • Further, the discussion on page 12 suggests that polycations disrupt cell walls and membranes. This should be discussed in more detail, particularly, because hydrophilic polycations are not disrupting cell membranes, but rather cross them without causing damage. The damages found in electron microscopy on dried samples are usually either artefacts or damages induced by the cells themselves indirectly caused by the biocidal action of the polymers, e.g., cell wall damage induced by inhibition of the teichoic acids for Gram-positive bacteria.

Response        We are aware of the potential problems caused during sample preparation for microscopy and have accounted for this by comparing PHMG-Cl treated and non-treated samples directly. We had added some comments relating to the effect of polycatonic polymers as part of above and modified our comment at the end of Pg. 12 to say that as a result of PHMG-Cl activity, cell walls and membranes were ‘affected’.

All the changes marked in red.

Round 2

Reviewer 1 Report

Serious problems that concern the methodological approach and the data interpretation is the possibility that throughout the study on eDNA there is possibility that the isolated DNA probe is a mix of eDNA and intracellular (i) DNA contributed from bacterial biofilm cells altered through the treatments. In support of this uncertainty is the result with the chromosomal gene for 16SrRNA which is present in most of the extracellular samples, incl. the "water only control", as well as the predominant red staining with PI of the "water-only control samples", indicating the predominance of dead bacterial cells in the biofilm that is presented as control.

For more details see attachment.

Author Response

  • The authors have their motivation in the possibility for horisontal gene transfer via the extracellular matrix of biofilms, and the possibility to block this by the selected disinfectant. The serious question regarding this MS is whether theevidence provided by the authors refers only to eDNA from the biofilms they cultivate, or in some of the samples, if not most, there is sufficient contribution from intracellular DNA (iDNA) as a result of disintegration of bacterial cells during the treatments. This question arises, e.g., from the results on the 16SrRNA gene amplification, Figure 3.

Response           The issue of ‘mixing’ eDNA with intracellular (iDNA) appears to be the core to this Reviewer’s concerns about our manuscript. We recognise the points being made and have addressed this in several sections of our revised manuscript where we make it clear that we are not concerned about the distinction between ‘old’ eDNA that may have accumulated throughout the growth of biofilms and ‘new’ eDNA that was released from our biofilm samples, as a result of partial dehydration to form a model for poorly cleaned biofilm residues that might be found in hospitals, and the action of biocides on these residues.

In our revised manuscript, we explicitly state that the DNA we are recovering from our model biofilms after biocidal treatment is a combination of the eDNA that had accumulated during the growth of the biofilm and the DNA (intracellular) freshly released by cells damaged by the biocide. We have made several additions to the revised manuscript to highlight this important point, as well as the fact that eDNA is considered to consist of DNA secreted by living cells and DNA released by lysis of dying cells (citing the recent review by Sivalingham et al. (2020). We have also added a section to the Introduction making the point that biocides might add to the accumulation of eDNA in biofilm residues as the result of poor or partial cleaning.

  • The authors use a chromosomal gene as a marker of eDNA. Bearing in mind the chromosomal location of this gene in bacteria, such DNA could be contributed to the extracellular matrix of the biofilm only in case the cell integrity is altered and the cell envelope - greatly permeabilized. This might happen to some extent or another during the whole life-span of the biofilms, however to estimate the contribution of such event, a control from fresh untreated biofilm would provide much tronger evidence in order to see whether this chromosomal gene has been present in the native biofilm, or actually it occurred in the "eDNA" of the "water-only control" as a result of the osmotic stress. This question arises, e.g., from the results on the 16SrRNA gene amplification, Figure 3. The presence of 16SrDNA in all samples except these treated with PHMG-Cl (Figure 3) could be interpreted, alternatively to the author's preferences, as evidence for the contribution of iDNA to their samples. This is a problem if only the eDNA is in the focus of the MS. To prove that they are examining eDNA per se, and not a intracellular DNA from broken cells, the authors need a more solid evidence than just their "water only" control.

Response           Both P. aeruginosa and S. aureus strains are regularly isolated from fresh water (including drinking water) indicating that they are resistant to hypo-osmotic stress and can survive under these conditions for far longer that the time we treated biofilms with biocide or the water-only controls. We have cited Mena and Gerba (2009) and Santo et al. (2020) in support of this in the revised manuscript.

We accept that the ‘water-only’ treatment of our biofilms with biocides as shown in Fig. 3 may have released some DNA because of osmotic stress or a result of physical disturbance during treatment and handling. However, these biofilms were already partially dehydrated to produce samples of our hospital biofilm-residue ‘model’ material in which we expect some cells to have been damaged or otherwise lysed to release the eDNA we were interested in ‘cleaning’. We recognise this the revised manuscript.

  • Another shorthcoming of Figure 3 is that the authors have not included a molecular marker.

Response           We did not run a sample of a DNA ladder with the gels shown in Fig. 3 as the size of the expected PCR amplicon was known (1,421 bp). We have added this to the figure legend for those readers who are interested

  • Another figure that arises confusion is Figure 6, especially with regards to "water-only" controls. The authors apply a modification of the live/dead staining approach. This would expectedly color live cells in green and dead cells plus eDNA in red. The principle of the method is that one of the stains (in this case SYBR green) would penetrate into the living cells, while the molecule of propidium iodide cannot penetrete through the unaltered cell envelope and can color only altered cells and eDNA. Of concern, in Figure 6 the control, "water-only treated" samples for both strains, and the "water-only treated" sample for the 5-day biofilm of the 5-day biofilm of aureus are all red, and do not seem to possess living, green-colored cells. This visual evidence once again puts in question the validity of the "water-only samples" as the only control of the experiments, under the applied experimental scheme.

 Response           On reflection, we recognise that Fig. 6 is confusing as in the PDF images we provided, the underlying SYBR Green signal used to visualise intracellular DNA was obscured by the dominant red PI signal used to visualise eDNA.

In the original images we provided, the PI eDNA (red) signal dominated the orthogonal and transect views and hid the SYBR Green intracellular DNA (green) signals. This is partially because both species are known to produce large amounts of eDNA, effectively coating or encasing cells, and partially due to the colour intensity settings we used for these images.

We have gone back to our CLSM experiments and selected a different set of images where the red and green signals were separated. For each of the PA and SA water-only and PHMG-Cl treatments, we now show the combined PI plus SYBR Green signal in an orthogonal view, and under that, the PI and then the SYBR Green signals separately using the same orthogonal view. We have also reported the signal ratios in the figure legend and rewritten the section of text presenting this work in the revised manuscript. The effect on PA and SA biofilms of PHMG-Cl differs, and we note this in the rewritten section of the Discussion too.

As both PA and SA are isolated from fresh and drinking water, we do not think it likely that the brief time the biofilm samples were exposed to deionised water in the ‘water-only’ treatment and in washing would have resulted in significant levels of cell lysis through hypo-osmotic stress. We have not reiterated this point and believe that the comments made at the start of the Results section and the Methods are sufficient.

  • As well, the authors avoided to comment on the recommendation to include detail on the sample preparation for CLSM. Neither in the reply to the referees, nor in the methodology of the corrected version, the authors have added details on the sample preparation, just the mode of coloring them and the microscopy approach. Mode of biofilm cultivation, mode of its transfer to the slide (if it was cultivated elsewhere), the mounting of the microscopic slides, buffers, pH, mounting medium, etc., could influence the final result.

Response. We have provided details of how biofilms were incubated and how they were assayed, including the staining used for CLSM. The use of these DNA strains and CLSM imaging is commonplace in biofilm research and has been extensively used for both P. aeruginosa and S. aureus. We accept that, in the case of the CLSM preparation, we did not indicate how biofilm samples were recovered from the glass vials but have added this detail to the Methods. Our staining of the samples for CLSM followed normal protocols. SYBR Green is provided in a buffer by the manufacturer. PI was made up in PBS. We have modified this section of the Methods to provide more details.

Round 3

Reviewer 1 Report

The results can be interpreted in two equally possible ways: (1)PHMG-Cl destroys eDNA while "water only" and other disinfectants do not; (2) All other disinfectants as well as the "water only" cause cell damage with the release of iDNA, while PHMG-Cl does not. For more details, please see the attached file.

Author Response

ok